# Intact synapse structure and function after combined knockout of PTPδ, PTPσ, and LAR

**Javier Emperador-Melero\*, Giovanni de Nola, Pascal S Kaeser\***

Department of Neurobiology, Harvard Medical School, Boston, United States

**Abstract** It has long been proposed that leukocyte common antigen-related receptor protein tyrosine phosphatases (LAR-RPTPs) are cell-adhesion proteins that control synapse assembly. Their synaptic nanoscale localization, however, is not established, and synapse fine structure after knockout of the three vertebrate LAR-RPTPs (PTPδ, PTPσ, and LAR) has not been tested. Here, superresolution microscopy reveals that PTPδ localizes to the synaptic cleft precisely apposed to postsynaptic scaffolds of excitatory and inhibitory synapses. We next assessed synapse structure in newly generated triple-conditional-knockout mice for PTPδ, PTPσ, and LAR, complementing a recent independent study of synapse function after LAR-RPTP ablation (Sclip and Südhof, 2020). While mild effects on synaptic vesicle clustering and active zone architecture were detected, synapse numbers and their overall structure were unaffected, membrane anchoring of the active zone persisted, and vesicle docking and release were normal. Hence, despite their localization at synaptic appositions, LAR-RPTPs are dispensable for presynapse structure and function.

**\*For correspondence:**
Javier_EmperadorMelero@hms.
harvard.edu (JE-M);
kaeser@hms.harvard.edu (PSK)

**Competing interests:** The authors declare that no competing interests exist.

## Introduction

Presynaptic nerve terminals are packed with neurotransmitter-laden vesicles that fuse at the active zone, membrane-attached protein machinery that forms vesicular release sites. Work over the past two decades has established that the unique synaptic architecture with nanoscale apposition of these secretory hotspots with receptors on postsynaptic cells allows for robust signal transmission (*Biederer et al., 2017*; *Südhof, 2012*). The assembly mechanisms of these transcellular molecular machines, however, remain largely obscure (*Emperador-Melero and Kaeser, 2020*; *Rizalar et al., 2021*; *Südhof, 2018*).

Leukocyte common antigen-related receptor protein tyrosine phosphatases (LAR-RPTPs) are transmembrane proteins often regarded as presynaptic organizers. Three LAR-RPTPs – PTPδ, PTPσ, and LAR – are expressed in the brain, bind to the active zone scaffolding protein Liprin-α and to synaptic cell-adhesion proteins, and recruit presynaptic material in artificial synapse formation assays (*Bomkamp et al., 2019*; *Han et al., 2018*; *Han et al., 2020a*; *Johnson and Van Vactor, 2003*; *Kwon et al., 2010*; *Pulido et al., 1995*; *Serra-Pagès et al., 1998*; *Takahashi et al., 2011*; *Yim et al., 2013*). While these data suggest roles in presynaptic assembly (*Fukai and Yoshida, 2020*; *Takahashi and Craig, 2013*; *Um and Ko, 2013*), LAR-RPTP localization and function at neuronal synapses are less clear. In invertebrates, loss-of-function mutations in LAR-RPTPs resulted in defects in axon guidance, increased active zone and synapse areas, and impaired neurotransmitter secretion (*Ackley et al., 2005*; *Clandinin et al., 2001*; *Desai et al., 1997*; *Kaufmann et al., 2002*; *Krueger et al., 1996*). In mice, RNAi-mediated knockdown of LAR-RPTPs or deletion of PTPσ caused generalized loss of synapse markers and defective synaptic transmission (*Dunah et al., 2005*; *Han et al., 2018*; *Han et al., 2020a*; *Han et al., 2020b*), leading to the model that LAR-RPTPs control synapse formation. Furthermore, mild synaptic and behavioral defects were observed in single gene constitutive knockouts (*Elchebly et al., 1999*; *Horn et al., 2012*; *Park et al., 2020*;

*Uetani et al., 2000*; *Wallace et al., 1999*). Contrasting the RNAi-based analyses, however, a recent study used conditional mouse gene targeting to ablate PTPδ, PTPσ, and LAR, and found no overt defects in neurotransmitter release (*Sclip and Südhof, 2020*), thereby questioning the general role of LAR-RPTPs in synapse assembly.

The lack of knowledge of LAR-RPTP nanoscale localization and of a characterization of vertebrate synapse structure after ablation of all LAR-RPTPs obscures our understanding of their roles as synapse organizers. Here, we establish that PTPδ is apposed to postsynaptic scaffolds of inhibitory and excitatory synapses using stimulated emission depletion (STED) microscopy, supporting that these proteins could control synapse formation or regulate synapse function. However, analyses of active zone protein composition, synapse ultrastructure, and synaptic transmission in newly generated conditional PTPδ/PTPσ/LAR triple-knockout mice reveal that these proteins are largely dispensable for synapse structure and function.

## Results

PTPδ, PTPσ, and LAR are encoded by *Ptprd, Ptprs*, and *Ptprf*, respectively. Conditional knockout mice for each gene were generated using homologous recombination (*Figure 1—figure supplement 1*). Alleles for PTPδ (*Farhy-Tselnicker et al., 2017*; *Sclip and Südhof, 2020*) and PTPσ (*Bunin et al., 2015*; *Sclip and Südhof, 2020*) were identical to previously reported alleles, while the LAR allele was newly generated. The floxed alleles for each gene did not impair survival and RPTP protein expression was readily detected (*Figure 1—figure supplement 1*). We intercrossed these alleles to generate triple-conditional knockout mice. In cultured hippocampal neurons, Cre recombinase was delivered at DIV6 by lentiviruses and expressed from a Synapsin promotor (*Liu et al., 2014*) and resulted in removal of PTPδ, PTPσ, and LAR, generating cTKO[RPTP] neurons (*Figure 1A,B*). Control[RPTP] neurons were obtained using an inactive version of Cre.

We first aimed at resolving the subsynaptic localization of LAR-RPTPs using STED microscopy. PTPδ antibody specificity was established using cTKO[RPTP] neurons as negative controls, while antibodies suitable for superresolution analyses of PTPσ or LAR could not be identified. To determine PTPδ localization, we selected side-view synapses with bar-like postsynaptic receptor scaffolds (PSD-95 and Gephyrin for excitatory and inhibitory synapses, respectively) on one side of a Synaptophysin-labeled nerve terminal (*Figure 1—figure supplement 2*, *Emperador-Melero et al., 2020*; *Held et al., 2020*; *Wong et al., 2018*). PTPδ, detected with antibodies against the extracellular fibronectin domains (*Shishikura et al., 2016*), was concentrated apposed to PSD-95 and Gephyrin, respectively, at distances of $24 \pm 17$ nm (PSD-95) and $28 \pm 11$ nm (Gephyrin) (*Figure 1D–I*). Only background signal typical for quantification of raw images (*Emperador-Melero et al., 2020*; *Held et al., 2020*; *Wang et al., 2016*; *Wong et al., 2018*) remained in cTKO[RPTP] neurons in STED (*Figure 1D–I*) and confocal (*Figure 1—figure supplement 3*) microscopy. This establishes that the extracellular portion of PTPδ localizes to the synaptic cleft. Given the presynaptic roles of LAR-RPTPs at invertebrate synapses and in synapse formation assays (*Ackley et al., 2005*; *Kaufmann et al., 2002*), the interactions with the active zone protein Liprin-α (*Pulido et al., 1995*; *Serra-Pagès et al., 1998*; *Wong et al., 2018*), and the asymmetry of the average STED side-view profile of PTPδ with a bias toward the presynapse (*Figure 1F,I*), it is likely that most PTPδ is presynaptic and localized at the active zone, but postsynaptic components cannot be excluded. Furthermore, most synapses contain PTPδ, as 88% of excitatory and 92% of inhibitory synapses had PTPδ peak intensities higher than three standard deviations above the average of the cTKO[RPTP] signal.

The subsynaptic PTPδ localization and its presence at most synapses is consistent with general roles of LAR-RPTPs in synapse organization. However, the synapse density, measured as Synaptophysin puncta, was unchanged in cTKO[RPTP] neurons (*Figure 1L–O*). Small increases in puncta intensity and area were detected (*Figure 1L–O*), consistent with enlargements observed in invertebrates (*Ackley et al., 2005*; *Kaufmann et al., 2002*). A recent independent study that also ablated LAR-RPTPs using mouse genetics found normal synapse densities as well (*Sclip and Südhof, 2020*). Together, these data challenge the model that LAR-RPTPs are master synapse organizers (*Dunah et al., 2005*; *Fukai and Yoshida, 2020*; *Han et al., 2018*; *Han et al., 2020a*; *Han et al., 2020b*; *Kwon et al., 2010*; *Takahashi and Craig, 2013*; *Um and Ko, 2013*; *Yim et al., 2013*). It

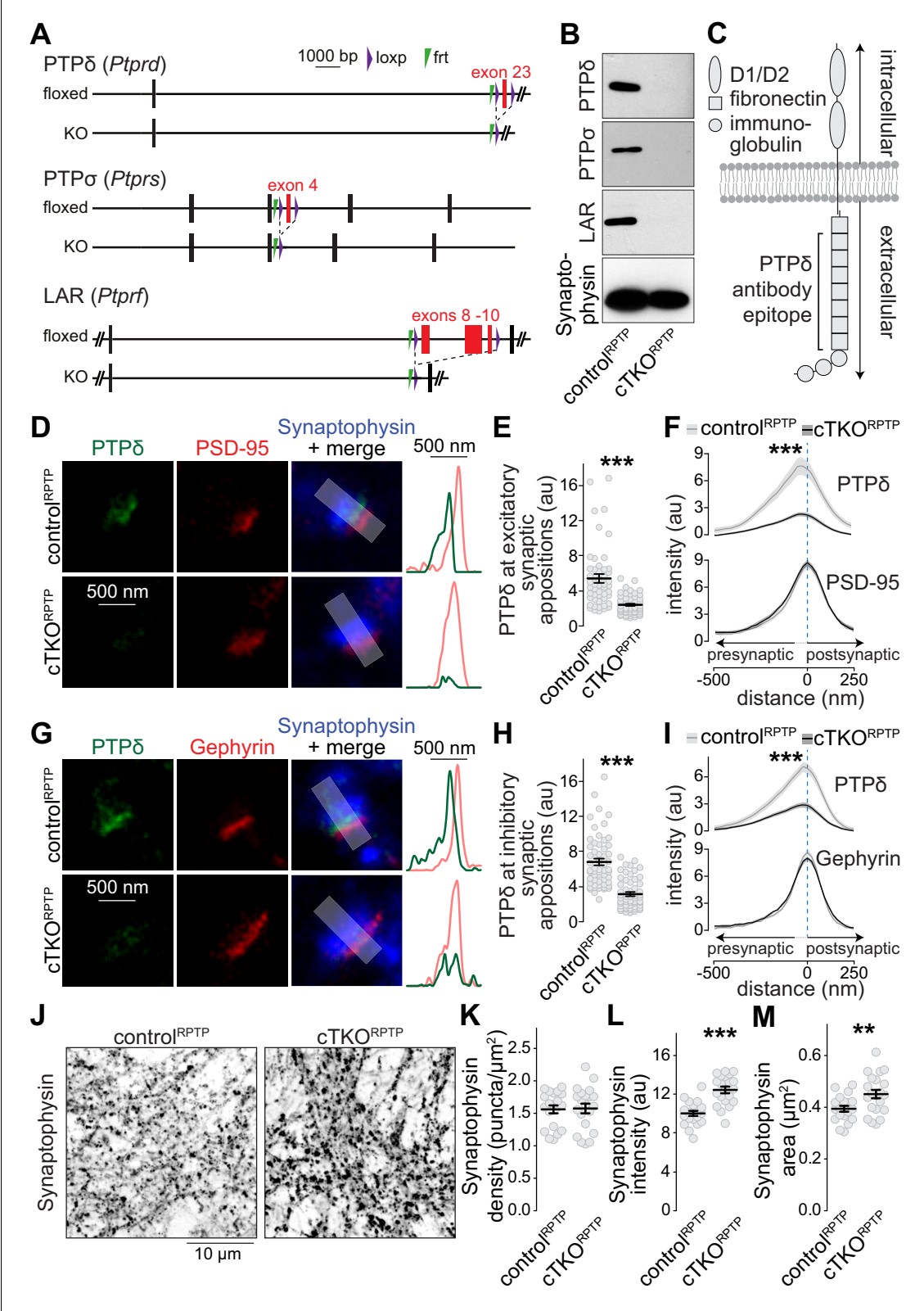

**Figure 1.** Nanoscale localization of PTPδ and assessment of synapse density in LAR-RPTP conditional triple-knockout neurons. (**A**) Diagram for simultaneous conditional knockout of PTPδ, PTPσ and LAR. (**B**) Example western blot of cultured neurons from PTPδ, PTPσ, and LAR triple floxed mice expressing Cre recombinase (to generate cTKO^RPTP neurons) or truncated Cre (to generate control^RPTP neurons). The bands detected in the cultured neurons correspond to the lower bands detected in brain homogenate shown in *Figure 1—figure supplement 1C*. (**C**) Diagram showing the general

*Figure 1 continued on next page*

*Figure 1 continued*

structure of LAR-RPTPs and the antibody recognition site for PTPδ (antibodies were generated using a peptide containing fibronectin domains 2, 3 and 8 *Shishikura et al., 2016*). (**D–F**) STED images (**D**), quantification of the peak intensity of PTPδ (**E**, STED) and average intensity profiles for PTPδ (STED) and PSD-95 (**F**, STED) at excitatory synapses. Side-view synapses were identified by the presence of bar-like PSD-95 signals (STED) at the edge of the vesicle cloud marked by Synaptophysin (confocal). Intensity profiles of shaded areas in the overlap images were used to determine the peak intensity of the protein of interest, and are shown on the right of the corresponding image. N (control^RPTP) = 50 synapses/3 cultures, N (cTKO^RPTP) = 54/3. (**G–I**) Same as D-F, but for inhibitory synapses identified by Gephyrin. N (control^RPTP) = 58/3 cultures, N (cTKO^RPTP) = 59/3. (**J–M**) Confocal images of cultured neurons stained with anti-Synaptophysin antibodies (**J**) and quantification of Synaptophysin puncta density (**K**), intensity (**L**) and size (**M**) detected using automatic two-dimensional segmentation. N (control^RPTP) = 20 images/3 independent cultures; N (cTKO^RPTP) = 21/3. The Synaptophysin confocal data are from the experiments shown in D-I. Data are plotted as mean ± SEM and were analyzed using two-way ANOVA tests (F, I, genotype *** for PTPδ), t-tests (E, L, M) or Mann-Whitney rank sum tests (H, K). **p<0.01, ***p<0.001.

The online version of this article includes the following figure supplement(s) for figure 1:

**Figure supplement 1.** LAR-RPTP conditional knockout mice.
**Figure supplement 2.** STED analysis workflow.
**Figure supplement 3.** Confocal analyses of PTPδ.

remains possible that LAR-RPTPs control assembly of a specific subset of synapses, which may explain why PTPδ ablation causes modest layer-specific impairments of synaptic strength (*Park et al., 2020*).

We next examined whether LAR-RPTPs have specific roles in presynaptic nanoscale structure. Electron microscopy of high-pressure frozen neurons (*Figure 2A–E*) revealed that synaptic vesicles were efficiently clustered at cTKO^RPTP synapses. A ~15% increase in the total synaptic vesicle number per synapse profile was detected, matching the modestly increased Synaptophysin signals (*Figure 1*) and the enhanced presence of vesicular markers in *C. elegans* mutants (*Ackley et al., 2005*). Notably, no differences in vesicle docking (defined as vesicles for which the electron dense membrane merges with the density of the target membrane) were observed. Synapse width, measured as the distance over which the pre- and postsynapse are apposed to one another and separated by a synaptic cleft, was increased by ~30%, again matching invertebrate phenotypes (*Kaufmann et al., 2002*). These data establish that LAR-RPTP ablation does not strongly impair synapse ultrastructure. LAR-RPTPs may shape aspects of the synaptic cleft, consistent with their localization and transsynaptic interactions and possibly similar to other synaptic cell-adhesion proteins, for example SynCAMs (*Perez de Arce et al., 2015*).

We assessed whether active zone proteins, which are present at normal levels in western blots after LAR-RPTP ablation (*Sclip and Südhof, 2020*), are anchored at the presynaptic membrane by LAR-RPTPs. STED microscopy was used to measure localization and peak levels of active zone proteins at excitatory (*Figure 2F–I*) and inhibitory (*Figure 2J–M*) synapses. RIM, Munc13-1, Ca$_V$2.1, and Liprin-α3 were localized within 30–60 nm of the postsynaptic scaffolds in control^RPTP and cTKO^RPTP synapses, as expected for these proteins (*Held et al., 2020*; *Wong et al., 2018*). Overall, there were no strong changes in their levels, but small increases in RIM and small decreases in Liprin-α3 were detected in both types of cTKO^RPTP synapses either by STED (*Figure 2F–M*) or confocal (*Figure 2— figure supplement 1*) microscopy. While binding between Liprin-α and LAR-RPTPs (*Pulido et al., 1995*; *Serra-Pagès et al., 1998*) may explain Liprin-α3 reductions, these data establish that other pathways are sufficient to recruit most Liprin-α3 to active zones. The higher levels of RIM may be compensatory to reductions in Liprin-α3 and could be related to the liquid–liquid phase separation properties of both proteins (*Emperador-Melero et al., 2020*; *McDonald et al., 2020*; *Wu et al., 2019*). Overall, we conclude that the active zone remains assembled and anchored to the target membrane in the absence of LAR-RPTPs.

A previous study found that LAR-RPTP ablation induced no strong defects in glutamate release, but regulated NMDARs through a transsynaptic mechanism (*Sclip and Südhof, 2020*). These findings are consistent with the near-normal synaptic ultrastructure and active zone assembly (*Figure 2*). We complemented this recent study by whole-cell recordings of inhibitory postsynaptic currents (IPSCs, *Figure 3*). Release evoked by single action potentials was similar between control^RPTP and cTKO^RPTP neurons, and IPSC kinetics were unaffected. The IPSC ratio of two consecutive stimuli (paired pulse ratio), which is inversely proportional to vesicular release probability (*Zucker and*

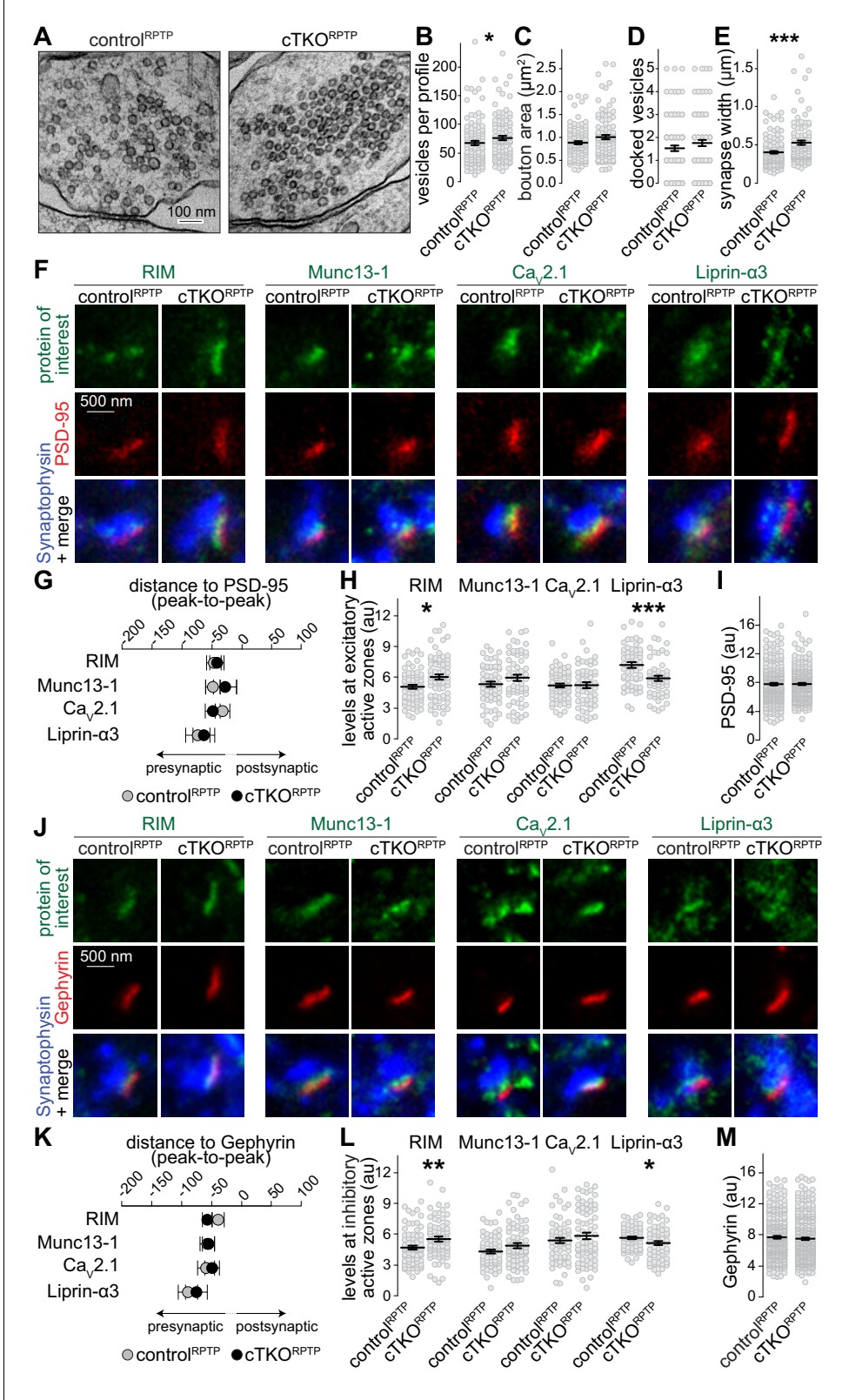

**Figure 2.** Synapse ultrastructure and active zone composition after LAR-RPTP triple knockout. (**A–E**) Electron micrographs (**A**) and quantification of the total number of vesicles per profile (**B**), bouton area (**C**), number of docked vesicles (**D**), and synapse width (**E**) assessed in single sections of high-pressure frozen neurons. N (control^RPTP) = 106 synapses/2 independent cultures, N (cTKO^RPTP) = 101/2. (**F–H**) STED example images of excitatory side-view synapses (**F**) and quantification of the distance to PSD-95 (**G**) and of peak intensities (**H**) of RIM, Munc13-1, Ca_V2.1, and Liprin-α3. RIM: N

*Figure 2 continued on next page*

*Figure 2 continued*

(control[RPTP]) = 68 synapses/3 independent cultures, N (cTKO[RPTP]) = 68/3; Munc13-1: N (control[RPTP]) = 57/3, N (cTKO[RPTP]) = 60/3; Ca$_V$2.1: N (control[RPTP]) = 64/3, N (cTKO[RPTP]) = 58/3; Liprin-α3: N (control[RPTP]) = 56/3, N (cTKO[RPTP]) = 53/3. (I) Quantification of the peak intensity of PSD-95. N (control[RPTP]) = 295/3; N (cTKO[RPTP]) = 293/3. (J–L) Same as (F–H), but for Gephyrin-containing inhibitory synapses. RIM: N (control[RPTP]) = 75/3 cultures, N (cTKO[RPTP]) = 79/3; Munc13-1: N (control[RPTP]) = 65/3, N (cTKO[RPTP]) = 72/3; Ca$_V$2.1: N (control[RPTP]) = 64/3, N (cTKO[RPTP]) = 71/3; Liprin-α3: N (control[RPTP]) = 65/3, N (cTKO[RPTP]) = 61/3. (M) Quantification of the peak intensity of Gephyrin. N (control[RPTP]) = 327/3; N (cTKO[RPTP]) = 342/3. Data are plotted as mean ± SEM and were analyzed using Mann–Whitney rank sum tests. *p<0.05, **p<0.01, ***p<0.001.

The online version of this article includes the following figure supplement(s) for figure 2:

**Figure supplement 1.** Confocal analyses of synaptic protein levels after ablation of LAR-RPTPs.

*Regehr, 2002*), was also unaffected. We conclude that synaptic vesicle exocytosis, here monitored via IPSCs, is not impaired by LAR-RPTP knockout.

## Discussion

Overall, we demonstrate that ablation of LAR-RPTPs from hippocampal neurons does not alter synapse density, synaptic vesicle docking, membrane anchoring of active zones, and synaptic vesicle release. This aligns with a parallel study that reported no loss of synaptic puncta and efficient release at excitatory synapses in cultured hippocampal neurons and in acute hippocampal brain slices

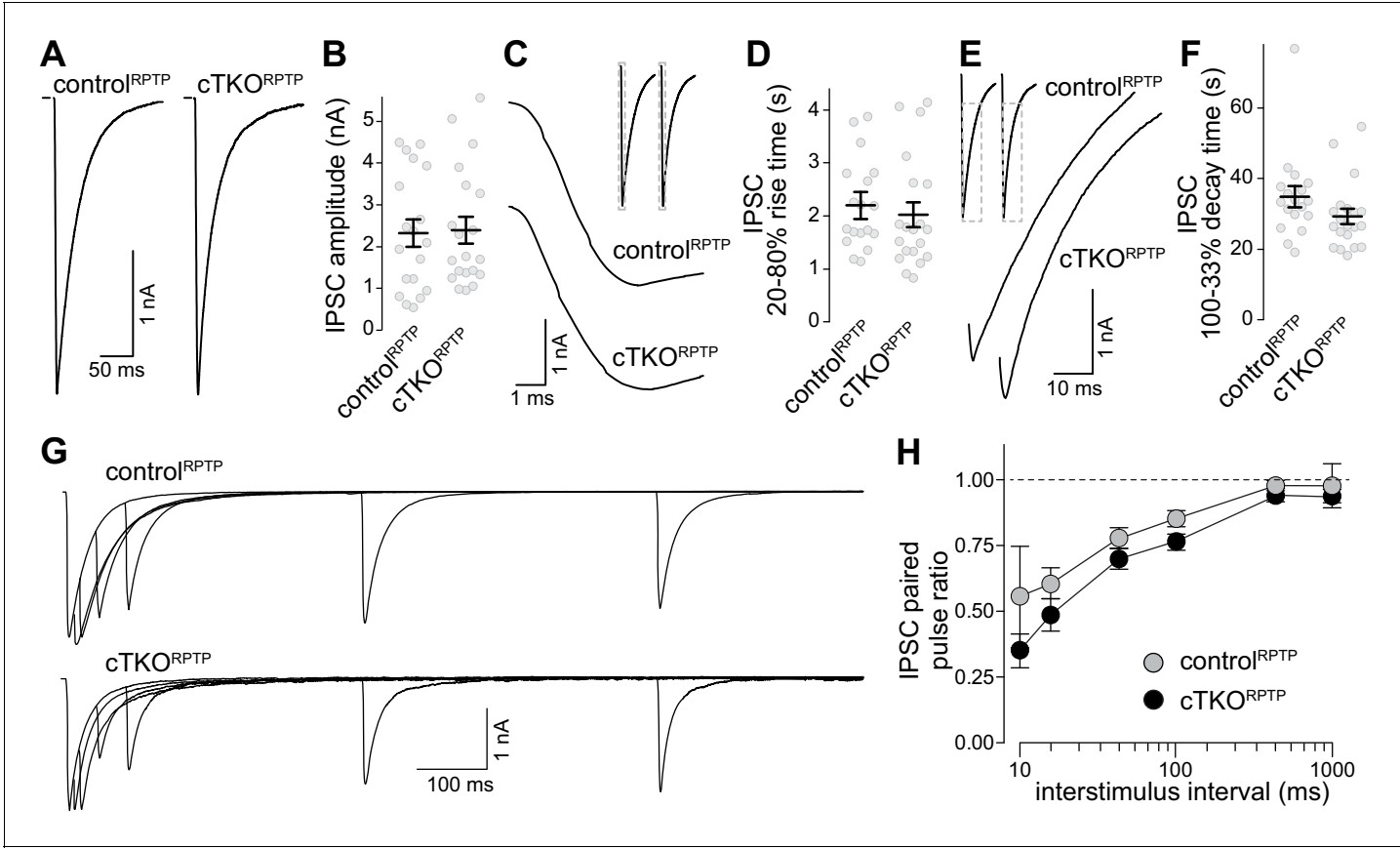

**Figure 3.** Inhibitory synaptic transmission in LAR-RPTP triple-knockout neurons. (A, B) Example traces (A) and average amplitudes (B) of single action potential evoked IPSCs. N (control[RPTP]) = 19 cells/3 independent cultures, N (cTKO[RPTP]) = 20/3. (C, D) Example zoomed-in traces of the IPSC rise (C) and quantification of 20–80% rise times (D) of evoked IPSCs, N as in (A, B). (E, F) Example zoomed-in traces of the IPSC decay (E) and quantification of 100–33% decay times (F) of evoked IPSCs. N as in (A, B). (G, H) Example traces (G) and average IPSC paired pulse ratios (H) at various interstimulus intervals. N (control[RPTP]) = 18/3, N (cTKO[RPTP]) = 19/3. Data are plotted as mean ± SEM and were analyzed using Mann–Whitney rank sum tests (B, D, F) or a two-way ANOVA (H), no significant differences were detected.

(*Sclip and Südhof, 2020*) upon LAR-RPTP knockout, but contrasts RNAi-based studies that led to models in which these RPTPs are major synapse organizers (*Dunah et al., 2005*; *Fukai and Yoshida, 2020*; *Han et al., 2018*; *Han et al., 2020a*; *Han et al., 2020b*; *Kwon et al., 2010*; *Takahashi and Craig, 2013*; *Um and Ko, 2013*; *Yim et al., 2013*). LAR-RPTPs belong to the superfamily of RPTPs (*Johnson and Van Vactor, 2003*), and it is possible that other RPTPs compensate for their loss. We note, however, that the time course of deletion in our knockout experiments is similar to the time course that is used in most RNAi-knockdown studies, and is hence unlikely to explain the differences. Other contributing factors could be different experimental preparations and off-target effects of knockdowns, which may generate artifacts in synapse formation experiments (*Südhof, 2018*). Altogether, we conclude that, while biochemical and synapse formation assays support synaptogenic activities for these proteins, synapses persist upon LAR-RPTP ablation, and their structure and function do not necessitate these proteins.

Our study establishes specific localization of PTPδ extracellular domains to the synaptic cleft. Hence, PTPδ is correctly positioned to locally execute synaptic functions, for example for shaping cleft geometry, to modulate presynaptic plasticity, or to control postsynaptic receptors (*Biederer et al., 2017*; *Sclip and Südhof, 2020*; *Uetani et al., 2000*). Such functions would not be at odds with the at most mild structural and functional effects after LAR-RPTP ablation, nor with upstream functions in neurite outgrowth and axon targeting (*Ackley et al., 2005*; *Clandinin et al., 2001*; *Desai et al., 1997*; *Krueger et al., 1996*; *Prakash et al., 2009*; *Shishikura et al., 2016*). Mechanisms of active zone anchoring to the target membrane, however, remain unresolved. Deletion of the major candidates, $Ca_V2$ channels (*Held et al., 2020*), Neurexins (*Chen et al., 2017*), and now LAR-PTPs (*Figures 1* and *2*), produces no major structural defects, indicating that active zones are most likely anchored to the plasma membrane through multiple parallel pathways that may or may not include these proteins (*Emperador-Melero and Kaeser, 2020*). Synaptic cell-adhesion proteins that contribute to synapse formation and function, for example SynCAMs, EphBs, Cadherins, Teneurins, and FLRTs, are plausible candidates that could act on their own or in concert with other proteins, including LAR-RPTPs, to contribute to active zone membrane anchoring (*Biederer et al., 2017*; *Südhof, 2018*).

# Materials and methods

## Key resources table

| Reagent type (species) or resource | Designation | Source or reference | Identifiers | Additional information |
|---|---|---|---|---|
| Genetic reagent (mouse) | Ptprd[tm2a(KOMP)Wtsi] | Acquired as frozen embryos from the Welcome Trust Sanger Institute, same allele as in *Farhy-Tselnicker et al., 2017*; *Sclip and Südhof, 2020* | Clone EPD0581_9_D04, MGI:4458607, RRID: IMSR_EM:11805 | |
| Genetic reagent (mouse) | C57BL/6N-Ptprs[tm1a(KOMP)Mbp]/Tcp | Acquired as frozen sperm from the Canadian Mouse Mutant Repository at the Hospital for Sick Children, same allele as in *Bunin et al., 2015*; *Sclip and Südhof, 2020* | Clone DEPD00535_1_D11; MGI:4840831, RRID:IMSR_CMMR:ABCA | |
| Genetic reagent (mouse) | Ptprf[tm1a(EUCOMM)Wtsi] | Acquired as embryonic stem cells from the Helmholtz Zentrum München | Clone EPD0697_1_D03; MGI:4887720, JAX: 637737 | |
| Recombinant DNA reagent | pFSW-CRE-EGFP | *Liu et al., 2014* | pHN131014 | |
| Recombinant DNA reagent | pFSW-deltaCRE-EGFP | *Liu et al., 2014* | pHN131015 | |
| Antibody | Goat anti-PTPσ | Thermo Fisher Scientific | RRID: AB_2607944, A114 | WB (1:200) |
| Antibody | Rat anti-PTPδ | Gift of Dr. F. Nakamura *Shishikura et al., 2016* | A229 | WB and ICC (1:500) |

*Continued on next page*

*Continued*

| Reagent type (species) or resource | Designation | Source or reference | Identifiers | Additional information |
|---|---|---|---|---|
| Antibody | Mouse anti-LAR | Clone E9B9S from Cell signaling | A156 | WB (1:500) |
| Antibody | Rabbit anti-Liprin-α3 | *Emperador-Melero et al., 2020* | A232 | ICC (1:250) |
| Antibody | Rabbit anti-RIM | SySy | RRID: AB_887774, A58 | ICC (1:500) |
| Antibody | Mouse anti-PSD-95 | NeuroMab | RRID: AB_10698024, A149 | ICC (1:500) |
| Antibody | Mouse anti-Gephyrin | SySy | RRID:AB_2232546, A8 | ICC (1:500) |
| Antibody | Guinea pig anti-Synaptophysin | SySy | RRID: AB_1210382, A106 | ICC (1:500) |
| Antibody | Rabbit anti-Ca$_V$2.1 | SySy | RRID: AB_2619841, A46 | ICC (1:500) |
| Antibody | Rabbit anti-Munc13-1 | SySy | RRID: AB_887733, A72 | ICC (1:500) |
| Antibody | Goat anti-rabbit Alexa Fluor 488 | Thermo Fisher | RRID: AB_2576217, S5 | ICC (1:250) |
| Antibody | Goat anti-mouse IgG1 Alexa Fluor 555 | Thermo Fisher | RRID: AB_2535769, S19 | ICC (1:250) |
| Antibody | Goat anti-mouse IgG2a Alexa Fluor 633 | Thermo Fisher | RRID: AB_1500826, S30 | ICC (1:250) |
| Antibody | Goat anti-rat IgG Alexa Fluor 488 | Thermo Fisher | RRID: AB_2534074, S55 | ICC (1:250) |
| Antibody | Goat anti-guinea pig IgG Alexa Fluor 405 | Abcam | RRID: AB_2827755, S51 | ICC (1:250) |
| Software | MATLAB | MathWorks | RRID: SCR_001622; https://www.mathworks.com/products/matlab.html | |
| Algorithm | Custom MATLAB code | *Liu et al., 2018* | https://github.com/hmslcl/3D_SIM_analysis_HMS_Kaeser-lab_CL and https://github.com/kaeserlab/3DSIM_Analysis_CL | |
| Software | Prism8 | GraphPad | RRID: SCR_002798, https://www.graphpad.com/scientific-software/prism | |
| Software | R | R Project | RRID: SCR_001905 | |
| Software | Fiji/ImageJ | NIH | RRID: SCR_002285 | |

## Mouse lines

PTPδ (*Ptprd*) mice were acquired as frozen embryos from the Welcome Trust Sanger Institute (Ptprd$^{tm2a(KOMP)Wtsi}$; clone EPD0581_9_D04, MGI:4458607, RRID:IMSR_EM:11805), and the same mutant allele was described in previous studies (*Farhy-Tselnicker et al., 2017*; *Sclip and Südhof, 2020*). PTPσ (*Ptprs*) mice were obtained as frozen sperm from the Canadian Mouse Mutant Repository at the Hospital for Sick Children (C57BL/6N-Ptprs$^{tm1a(KOMP)Mbp}$/Tcp; clone DEPD00535_1_D11; MGI:4840831, RRID:IMSR_CMMR:ABCA) and were also used previously (*Bunin et al., 2015*; *Sclip and Südhof, 2020*). Embryonic stem cells containing the LAR (*Ptprf*) mutant allele were obtained from the Helmholtz Zentrum München (Ptprf$^{tm1a(EUCOMM)Wtsi}$; clone EPD0697_1_D03; MGI:4887720). Mutant alleles were originally generated using homologous recombination by the international knockout consortium (*Bradley et al., 2012*; *Skarnes et al., 2011*). Frozen embryos (PTPδ), frozen sperm (PTPσ), or embryonic stem cells (LAR) were used to establish the respective mouse lines through the Transgenic Mouse Core (DF/HCC) at Harvard Medical School. For generation of the LAR mutant mice, the embryonic stem cells were expanded, the genotype was confirmed by PCR and sequencing, and injection into C57BL/6 blastocysts was used to generate chimeric founders. After germline transmission, the mice were crossed to Flp-expressing mice (*Dymecki, 1996*) to remove the LacZ and Neomycin cassettes to generate the conditional allele. The same crossing was performed with the cryo-recovered PTPδ and PTPσ mice. This strategy generated conditional 'floxed' alleles for each gene, in which exon 23 for *Ptprd*, exon 4 for *Ptprs*, and exons 8,

9, and 10 for *Ptprf* were flanked by loxP sites. Survival of each individual floxed allele was analyzed in offsprings of heterozygote matings through comparison of obtained genotypes of offsprings on or after P14 to expected genotypes for Mendelian inheritance. The three floxed lines were intercrossed and maintained as triple-homozygote mice. The conditional PTPδ, PTPσ, and LAR alleles were genotyped using the oligonucleotide primers CAGAGGTGGCTCATGTGC and GCCCAACCC TCAATTGTCAGAC (PTPδ, 465 and 287 bp bands for the floxed and wild-type alleles, respectively), GAGTCCTCAAACCAGGCCCTG and GGTGAGACCAGGGTGGGTTC (PTPσ, 522 and 345 bp bands for the floxed and wild-type alleles, respectively), and GATGGTCCCTCTGGAGAC and GCCAAGCCCATGCTCAGAG (LAR, 498 and 289 bp bands for the floxed and wild-type alleles, respectively). All animal experiments were approved by the Harvard University Animal Care and Use Committee.

## Neuronal cultures and production of lentiviruses

Primary hippocampal cultures were prepared as described (*Emperador-Melero et al., 2020*; *Held et al., 2020*; *Wong et al., 2018*). Briefly, hippocampi of newborn (postnatal days P0 or P1) pups were digested in papain, and neurons were plated onto glass coverslips in plating medium composed of mimimum essential medium (MEM) supplemented with 0.5% glucose, 0.02% NaHCO$_3$, 0.1 mg/ml transferrin, 10% fetal select bovine serum, 2 mM L-glutamine, and 25 mg/ml insulin. After 24 hr, plating medium was exchanged with growth medium composed of MEM with 0.5% glucose, 0.02% NaHCO$_3$, 0.1 mg/ml transferrin, 5% fetal select bovine serum (Atlas Biologicals), 2% B-27 supplement, and 0.5 mM L-glutamine. At DIV2–3, cytosine b-D-arabinofuranoside (AraC) was added to a final concentration of 1–2 mM. Cultures were kept in a 37°C incubator for a total of 14–16 d before analyses proceeded. Lentiviruses were produced in HEK293T cells maintained in DMEM supplemented with 10% bovine serum and 1% penicillin/streptomycin. HEK293T cells were transfected using the calcium phosphate method with a combination of three lentiviral packaging plasmids (REV, RRE, and VSV-G) and a separate plasmid encoding either Cre recombinase or inactive Cre, at a molar ratio of 1:1:1:1. Twenty-four hours after transfection, the medium was changed to neuronal growth medium, and 18–30 hr later, the supernatant was used for viral transduction. Neuronal cultures were infected 6 d after plating with lentiviruses expressing EGFP-tagged Cre recombinase (pHN131014) or an inactive variant (pHN131015) expressed under the human Synapsin promotor (*Liu et al., 2014*), and infection rates were assessed via nuclear EGFP fluorescence. Only cultures in which no non-infected neurons could be detected were used for analyses.

## Western blotting

Cell lysates were collected from DIV14–15 neuronal cultures in 1x sodium dodecyl sulfate (SDS) solution in PBS (diluted from 3x SDS sample buffer). For tissue collection, brains of postnatal days P21–P28 mice were homogenized using a glass-Teflon homogenizer in 5 ml of ice-cold homogenizing solution (150 mM NaCl, 25 mM HEPES, 4 mM EDTA, and 1% Triton X-100, pH 7.5), and 3x SDS sample buffer was added (to a final concentration of 1x). Samples were denatured at 100°C for 10 min, run on SDS–PAGE gels, and then transferred to nitrocellulose membranes for 6.5 hr at 4°C in buffer containing (per liter) 200 ml methanol, 14 g glycine, and 6 g Tris. Next, membranes were blocked for 1 hr at room temperature in TBST (Tris-buffered saline + 0.1% Tween-20), with 10% non-fat milk powder and 5% normal goat serum. Membranes were incubated with primary antibodies overnight at 4°C in TBST with 5% milk and 2.5% goat serum, followed by 1 hr incubation with horseradish peroxidase (HRP)-conjugated secondaries at room temperature. Three 5 min washes were performed between every step. Protein bands were visualized using chemiluminescence and exposure to film. The primary antibodies were as follows: goat anti-PTPσ (A114, 1:200, RRID: AB_2607944), rat anti-PTPδ (A229, 1:500, gift of Dr. F. Nakamura; *Shishikura et al., 2016*), mouse anti-LAR (A156, 1:500, clone E9B9S from Cell Signaling), and mouse anti-Synaptophysin (A100, 1: 5000, RRID: AB_887824). For PTPσ, normal goat serum was substituted by rabbit serum. The secondary antibodies were HRP-conjugated goat anti-mouse IgG (S44, 1:10,000, RRID:AB_2334540), HRP-conjugated goat anti-rabbit IgG (S45, 1:10,000, RRID:AB_2334589), HRP-conjugated goat anti-rat IgG (S46, 1:10,000, RRID: AB_10680316), and HRP-conjugated anti-goat antibodies (S60, 1:10,000).

## Immunofluorescence staining of neurons

Neurons grown on #1.5 glass coverslips were fixed at DIV15 in 4% paraformaldehyde (PFA) for 10 min (except for staining with anti-Ca$_V$2.1 and PTPδ antibodies, for which 2% PFA was used), followed by blocking and permeabilization in PBS containing 3% BSA/0.1% Triton X-100/PBS for 1 hr at room temperature. Incubation with primary and secondary antibodies was performed overnight at 4°C and for 1 hr at room temperature, respectively. Samples were post-fixed in 4% PFA for 10 min and mounted onto glass slides using ProLong diamond mounting medium. Antibodies were diluted in blocking solution. Three 5 min washes with PBS were performed between steps. Primary antibodies used were as follows: rabbit anti-Liprin-α3 (A232, 1:250; *Emperador-Melero et al., 2020*), rabbit anti-RIM (A58, 1:500, RRID: AB_887774), mouse anti-PSD-95 (A149, 1:500; RRID: AB_10698024), mouse anti-Gephyrin (A8, 1:500; RRID:AB_2232546), guinea pig anti-Synaptophysin (A106, 1:500; RRID: AB_1210382), rabbit anti-Munc13-1 (A72, 1:500; RRID: AB_887733), rat anti-PTPδ (A229; 1:500; gift of Dr. F. Nakamura; *Shishikura et al., 2016*), and rabbit anti-Ca$_V$2.1 (A46, 1:500; RRID: AB_2619841). Secondary antibodies used: goat anti-rabbit Alexa Fluor 488 (S5; 1:250, RRID:AB_2576217), goat anti-mouse IgG1 Alexa Fluor 555 (S19, 1:250, RRID: AB_2535769), goat anti-mouse IgG2a Alexa Fluor 633 (S30, 1:250, RRID: AB_1500826), goat anti-rat IgG Alexa Fluor 488 (S55, 1:250, RRID: AB_2534074) and goat anti-guinea pig IgG Alexa Fluor 405 (S51, 1:250, RRID: AB_2827755).

## STED and confocal imaging

All images were acquired as described (*Emperador-Melero et al., 2020*; *Held et al., 2020*; *Wong et al., 2018*) using a Leica SP8 Confocal/STED 3× microscope equipped with an oil-immersion 100 × 1.44 N.A objective, white lasers, gated detectors, and 592 nm and 660 and 770 nm depletion lasers. For every region of interest (ROI), quadruple color sequential confocal scans for Synaptophysin, PSD-95, Gephyrin, and a protein of interest (RIM, Munc13-1, PTPδ, Liprin-α or Ca$_V$2.1) were followed by triple-color sequential STED scans for PSD-95, Gephyrin, and the protein of interest. Synaptophysin was only imaged in confocal mode because of depletion laser limitations. Identical settings were applied to all samples within an experiment. For analyses of synapse density, Synaptophysin signals were used to generate ROIs using automatic detection with a size filter of 0.4–2 μm$^2$ (code available at https://github.com/kaeserlab/3DSIM_Analysis_CL and https://github.com/hmslcl/3D_SIM_analysis_HMS_Kaeser-lab_CL) and as described before (*Emperador-Melero et al., 2020*; *Held et al., 2020*; *Liu et al., 2018*). To measure synaptic levels of PTPδ, RIM, Munc13-1, Liprin-α3, and Ca$_V$2.1 in confocal mode, a mask was generated in ImageJ using an automatic threshold in the Synaptophysin or the PSD-95 channel, and the levels were measured within that mask. For STED quantification, side-view synapses were selected while blind to the protein of interest. They were defined as synapses that contained a vesicle cluster (imaged in confocal mode) with a single bar-like Gephyrin or PSD-95 structure (imaged by STED) along the edge of the vesicle cluster. A 1 μm long, 250 nm wide profile was selected perpendicular to the postsynaptic density marker and across its center. The peak levels of the protein of interest were then measured as the maximum intensity of the line profile within 100 nm of the postsynaptic density marker peaks (estimated area based on *Wong et al., 2018*) after applying a 5-pixel rolled average. For side-view plots, line scans from individual side-view synapses were aligned to the peak of PSD-95 or Gephyrin after the 5-pixel rolling average was applied, and averaged across images. Only for representative images, a smooth filter was added, brightness and contrast were linearly adjusted, and images were interpolated to match publication standards. These adjustments were made identically for images within an experiment. All quantitative analyses were performed on original images without any processing, and all data were acquired and analyzed by an experimenter blind to genotype. For PTPδ STED analyses, synapses were considered PTPδ positive if the peak intensity was higher than three standard deviations above the average of the cTKO$^{RPTP}$ signal, assessed separately in each individual culture.

## High-pressure freezing and electron microscopy

Electron microscopy was performed as previously described (*Held et al., 2020*; *Wang et al., 2016*). Briefly, DIV15 neurons grown on 6 mm sapphire cover slips were frozen with a Leica EM ICE high-pressure freezer in extracellular solution containing 140 mM NaCl, 5 mM KCl, 2 mM CaCl$_2$, 2 mM

MgCl$_2$, 10 mM glucose, 10 mM Hepes, 20 µM CNQX, 50 µM AP5, and 50 µM picrotoxin (pH 7.4, ~310 mOsm). Freeze substitution was done in acetone containing 1% osmium tetroxide, 1% glutaraldehyde, and 1% H$_2$O as follows: −90°C for 5 hr, 5°C per hour to −20°C, −20°C for 12 hr, and 10°C per hour to 20°C. Samples were then infiltrated in epoxy resin and baked at 60°C for 48 hr followed by 80°C overnight. Next, sapphire coverslips were removed from the resin block by heat shock, and samples were sectioned at 50 nm with a Leica EM UC7 ultramicrotome. Sections were mounted on a nickel slot grid with a carbon-coated formvar support film and counterstained by incubation with 2% lead acetate solution for 10 s, followed by rinsing with distilled water. Samples were imaged with a JEOL 1200EX transmission electron microscope equipped with an AMT 2 k CCD camera. Images were analyzed using SynapseEM, a MATLAB macro provided by Drs. M. Verhage and J. Broeke. Bouton area was measured by outlining the perimeter of each synapse profile. Docked vesicles were defined as vesicles touching the presynaptic plasma membrane opposed to the PSD, with the electron density of the vesicular membrane merging with that of the target membrane. Synapse width was measured as the area between synaptically apposed cells in which an evenly spaced cleft was present and associated with pre- and postsynaptic densities. All data were acquired and analyzed by an experimenter blind to the genotype.

## Electrophysiology

Electrophysiological recordings were performed as described before (*Emperador-Melero et al., 2020*; *Held et al., 2020*; *Wang et al., 2016*). Neurons were recorded at DIV15–16 in whole-cell patch-clamp configuration at room temperature in extracellular solution containing (in mM) 140 NaCl, 5 KCl, 1.5 CaCl$_2$, 2 MgCl$_2$, 10 HEPES (pH 7.4), and 10 Glucose, supplemented with 20 µM CNQX and 50 µM D-AP5 to block AMPA and NMDA receptors, respectively. Glass pipettes were pulled at 2.5–4 MΩ and filled with intracellular solution containing (in mM) 40 CsCl, 90 K-gluconate, 1.8 NaCl, 1.7 MgCl$_2$, 3.5 KCl, 0.05 EGTA, 10 HEPES, 2 MgATP, 0.4 Na$_2$-GTP, 10 phosphocreatine, CsOH, and 4 mM QX314-Cl (pH 7.4). Neurons were clamped at −70 mV, and series resistance was compensated to 4–5 MΩ, and recordings in which the uncompensated series resistance was >15 MΩ at any time during the experiment were discarded. Electrical stimulation was applied using a custom bipolar electrode made from Nichrome wire. A Multiclamp 700B amplifier and a Digidata 1550 digitizer were used for data acquisition, sampling at 10 kHz and filtering at 2 kHz. Data were analyzed using pClamp. The experimenter was blind during data acquisition and analyses.

## Statistics

Summary data are shown as mean ± SEM. Unless noted otherwise, significance was assessed using t-tests or Mann–Whitney U tests depending on whether assumptions of normality and homogeneity of variances were met (assessed using Shapiro or Levene's tests, respectively). Two-way ANOVA tests on a 200 nm wide window centered around the PSD-95 peak were used for line profile analyses of STED data, and chi-square tests were used to assess mouse survival ratios. All data were analyzed by an experimenter blind to the genotype. For all quantifications, the specific tests used are stated in the corresponding figure legends.

## Acknowledgements

We thank J Wang, M Han, and C Qiao for technical support, Dr. H Nyitrai for help and advice, Dr. F Nakamura for PTPδ antibodies, and Drs. M Verhage and J Broeke for the SynapseEM MATLAB macro. This work was supported by grants from the NIH (R01NS083898 and R01MH113349 to PSK), by Harvard Medical School (to PSK), and by a fellowship from the Alice and Joseph E Brooks postdoctoral fund (to JEM). For mutant mice, we thank the Welcome Trust Sanger Institute Mouse Genetics Project (Sanger MGP), its funders and INFRAFRONTIER/EMMA (http://www.infrafrontier.eu, *Ptprd*, funding information may be found at http://www.sanger.ac.uk/mouseportal and associated primary phenotypic information at http://www.mousephenotype.org), the Canadian Mouse Mutant Repository at the Hospital for Sick Children (*Ptprs*), and the Helmholtz Zentrum München (*Ptprf*). We thank the Dana-Farber/Harvard Cancer Center for the use of the Transgenic Mouse Core (in part supported by an NCI P30 Center Core Grant CA006516), which performed cryo-recovery, embryonic stem cell expansion, and blastocyst injections. We acknowledge the Neurobiology

Imaging Facility (supported by an NINDS P30 Center Core Grant NS072030), and the Electron Microscopy Facility at Harvard Medical School.

## Additional information

### Funding

| Funder | Grant reference number | Author |
|---|---|---|
| National Institute of Mental Health | R01MH113349 | Pascal S Kaeser |
| National Institute of Neurological Disorders and Stroke | R01NS083898 | Pascal S Kaeser |
| Harvard Medical School | | Pascal S Kaeser |
| Harvard Medical School | Brooks Fellowship | Javier Emperador-Melero |

The funders had no role in study design, data collection and interpretation, or the decision to submit the work for publication.

### Author contributions

Javier Emperador-Melero, Conceptualization, Formal analysis, Investigation, Methodology, Writing - original draft; Giovanni de Nola, Formal analysis, Investigation, Methodology; Pascal S Kaeser, Conceptualization, Formal analysis, Funding acquisition, Writing - original draft

### Author ORCIDs

Javier Emperador-Melero https://orcid.org/0000-0002-1364-4935
Pascal S Kaeser https://orcid.org/0000-0002-1558-1958

### Ethics

Animal experimentation: All animal experiments were performed according to institutional guidelines of Harvard University, and were in strict accordance with the recommendations in the Guide for the Care and Use of Laboratory Animals of the National Institutes of Health. The animals were handled according to protocols (protocol number IS00000049) approved by the institutional animal care and use committee (IACUC).

### Decision letter and Author response

Decision letter https://doi.org/10.7554/eLife.66638.sa1

## Additional files

### Supplementary files

• Transparent reporting form

### Data availability

Data generated or analysed during this study are included in the manuscript and supporting files.

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
