## [Decision Letter]

**Acceptance summary:**

LAR-RPTP proteins have received considerable attention as constituent synaptic proteins, yet their function remains elusive. This study used super-resolution microscopy, electron microscopy and synaptic electrophysiology to determine whether there was an effect on synaptic transmission and synapse structure after transgenic elimination of the three major proteins. Little-to-no phenotype was observed. The emerging picture is that these molecules are not necessary for synapse formation, stability or integrity. Together with a recent paper by Sclip and Südhof, (2020), these studies indicate that LAR-RTPs likely play other roles beyond synapse formation.

**Decision letter after peer review:**

Congratulations, we are pleased to inform you that your article, "Intact synapse structure and function after combined knockout of PTPδ, PTPσ and LAR", has been accepted for publication in eLife.

Reviewer #1:

This is a straightforward and well-executed study of the LAR-RPTP triple knockout. This work follows upon a recent paper from the Sudhof laboratory characterizing a similar triple knockout. Since these data were generated at nearly identical times, with great effort and expense, it is generally e*Life* policy not to consider the prior work as having scooped the authors (as I understand the original policies at e*Life*).

I have no additional requests for data or experiments, as any such speculation as to where a phenotype might arise would be pure speculation. In my opinion, the study stands on its own as a complement to the recent work from the Sudhof lab as part of a definitive synaptic characterization of these interesting synaptic proteins.

Reviewer #2:

In this straightforward study Emperador-Molero et al challenge the prevailing notion that the cell-adhesion proteins LAR-RTPs control synapse assembly. The authors provide strong evidence that synapses are normally formed and synaptic transmission is unaffected in triple cKO neurons. Only a subtle phenotype was observed. Using superresolution microscopy (STED), the authors also show that one of the LAR-RPTs PTPδ is presynaptic and localized at the active zone. Overall, the results are convincing and support the claims that are made. Together with a recent paper by Sclip and Südhof (2020), these studies indicate that LAR-RTPs likely play other roles beyond synapse formation.

Reviewer #3:

The manuscript "Intact synapse structure and function after combined knockout of PTPδ, PTPσ and LAR" by Emperador-Melero, de Nola, and Kaeser examines the impact of knocking out three members of the LAR-RPTPs. A number of studies have suggested that these proteins are required for synapse formation or assembly. Contrary to the previous findings that were based on single knockouts or shRNA-based approaches, these authors do not observe changes in synapse number or see other indications of large-scale synaptic loss. Using STED they conduct super-resolution analysis of the relationship and content of a number of synaptic proteins found at both excitatory and inhibitory synapses. They see defects in the organization of a protein known to associate with these proteins but only subtle differences relationship between the pre- and postsynaptic terminals. Given the number of studies suggesting an important function of LAR-RPTPs these findings will be of interest to many. One limitation of the study is that despite the presence of these proteins at many synapses and the subtle changes knocking them out makes to synapses, the authors do not identify another function for these molecules. Regardless, this is a carefully done study, and there are few technical concerns.